# A Lightweight Cherry Tomato Maturity Real-Time Detection Algorithm Based on Improved YOLOV5n

Congyue Wang [1,2] , Chaofeng Wang [1,2], Lele Wang [2], Jing Wang [1,2], Jiapeng Liao [1,2], Yuanhong Li [1,2,*] and Yubin Lan [1,2,3,4,*]

1   College of Electronic Engineering (College of Artificial Intelligence), South China Agricultural University, Guangzhou 510642, China
2   National Center for International Collaboration Research on Precision Agricultural Aviation Pesticides Spraying Technology, South China Agricultural University, Guangzhou 510642, China; shinylele@163.com
3   Guangdong Laboratory for Lingnan Modern Agriculture, South China Agricultural University, Guangzhou 510642, China
4   Department of Biological and Agricultural Engineering, Texas A&M University, College Station, TX 77843, USA
*   Correspondence: liyuanhong@scau.edu.cn (Y.L.); ylan@scau.edu.cn (Y.L.)

**Abstract:** To enhance the efficiency of mechanical automatic picking of cherry tomatoes in a precision agriculture environment, this study proposes an improved target detection algorithm based on YOLOv5n. The improvement steps are as follows: First, the K-means++ clustering algorithm is utilized to update the scale and aspect ratio of the anchor box, adapting it to the shape characteristics of cherry tomatoes. Secondly, the coordinate attention (CA) mechanism is introduced to expand the receptive field range and reduce interference from branches, dead leaves, and other backgrounds in the recognition of cherry tomato maturity. Next, the traditional loss function is replaced by the bounding box regression loss with dynamic focusing mechanism (WIoU) loss function. The outlier degree and dynamic nonmonotonic focusing mechanism are introduced to address the boundary box regression balance problem between high-quality and low-quality data. This research employs a self-built cherry tomato dataset to train the target detection algorithms before and after the improvements. Comparative experiments are conducted with YOLO series algorithms. The experimental results indicate that the improved model has achieved a 1.4% increase in both precision and recall compared to the previous model. It achieves an average accuracy mAP of 95.2%, an average detection time of 5.3 ms, and a weight file size of only 4.4 MB. These results demonstrate that the model fulfills the requirements for real-time detection and lightweight applications. It is highly suitable for deployment in embedded systems and mobile devices. The improved model presented in this paper enables real-time target recognition and maturity detection for cherry tomatoes. It provides rapid and accurate target recognition guidance for achieving mechanical automatic picking of cherry tomatoes.

**Keywords:** precision agriculture; YOLOv5; cherry tomato; maturity detection; CA; WIoU

## 1. Introduction

Cherry tomatoes, cherished by consumers for their richness in various nutrients, such as lycopene, organic acids, and vitamins, hold a significant place in daily life. Mechanical picking, as a crucial stage in fruit production, is still predominantly reliant on manual labor. Nevertheless, manual picking proves to be labor-intensive and expensive. By adopting mechanical picking, not only can costs be reduced and picking efficiency improved, but it can also contribute to enhancing the economic benefits of fruit farmers [1,2]. However, the growth characteristics of cherry tomatoes are intricate. The fruits exhibit diversity in terms of size, maturity, and location, often overlapping with one another. The dense presence of branches and leaves, coupled with complex lighting conditions, poses challenges for rapid

and accurate identification of cherry tomatoes during the harvesting process [3]. Furthermore, cherry tomatoes are susceptible to post-picking storage difficulties, and exhibit high perishability during transportation and sales. Consequently, it becomes necessary to pick cherry tomatoes of varying maturity levels based on specific transportation and storage requirements [4]. To effectively organize the labor force and pick at the opportune moment, obtaining precise positioning information of cherry tomatoes and accurate distribution details of different maturity levels quickly and accurately becomes an essential requirement for realizing mechanized picking. It stands as a crucial step in the implementation of precision agricultural technology [5,6].

In recent years, there has been a growing trend among domestic and international experts to integrate computer vision technology into the agricultural sector. The exploration of precise fruit recognition and fruit maturity classification algorithms has gained momentum, encompassing both conventional feature-based recognition methods and deep learning approaches like convolutional neural networks [7].

Traditional recognition methods based on digital image processing mainly match target fruits by extracting features such as color, geometric shape, and texture from the images. For example, Surya Prabha et al. [8] proposed a ripeness classification algorithm based on color and size, using first-order to fourth-order moments of the R, G, B components in banana images to identify the ripeness stages of bananas. Liu et al. [9] presented a color-based method for grapefruit classification and detection, transforming the image from RGB space to Y'CbCr space to recognize the ripeness of grapefruits. Lin et al. [10] used the Hough transform based on color and texture information to identify fruits like citrus and tomatoes using contour information. Bron et al. [11] introduced a chlorophyll fluorescence analysis technique for papaya ripeness grading. Kurtulmus et al. [12] proposed a citrus detection algorithm based on color images, using circular Gabor texture features and feature aggregation to successfully recognize green citrus. Although traditional recognition methods based on digital image processing have achieved some success in feature design, they have certain limitations. They primarily rely on fixed color features to match target fruits, and their adaptability to lighting variations and color shifts is limited, resulting in suboptimal performance in dealing with color instability [13]. Additionally, when dealing with complex background conditions, their reliance on geometric shape features such as edge detection and contour extraction leads to poor robustness in matching target fruits [14]. Similarly, texture feature extraction methods are often limited by manually designed filters and feature descriptors, which may cause computational inefficiency when processing large amounts of texture samples [15].

Traditional methods require manual feature design, leading to low detection efficiency, long processing times, and susceptibility to subjective factors, limiting their accuracy and robustness in complex scenes. To overcome these limitations, in recent years, deep learning techniques based on convolutional neural networks (CNNs) have become a research hotspot. Deep learning combines the processes of feature extraction, feature selection, and feature classification, with deeper structures and stronger learning capabilities, resulting in advantages such as efficiency, accuracy, and speed, making fruit recognition more intelligent and automated [16]. In related research, Gai et al. [17] proposed an improved YOLOv4 deep learning algorithm combined with DenseNet, capable of detecting three maturity stages of cherry fruits and performing well in the presence of overlapping and occluded fruits. Wang L et al. [18] introduced an improved lightweight YOLOv4-Tiny object detection algorithm, capable of recognizing and detecting three different maturity stages of blueberry fruits. Additionally, Wang et al. [19] improved the YOLOv5s algorithm by introducing the CBAM attention module to refine the effective feature information of lychee fruits with three densities and two maturity stages, achieving a mAP of 92.4%. Moreover, to address the recognition problem of unripe plums, Niu et al. [20] proposed a detection algorithm named YOLOv5-plum, and employed multi-scale training to achieve good detection performance for overlapping and occluded fruits. Furthermore, Chen et al. [21] used a combination of ResNet34 and YOLOv5 to identify and determine the three matu-

rity stages of citrus fruits with an accuracy of up to 95.07%. These studies demonstrate the potential of deep learning-based object detection algorithms in fruit recognition, and provide strong support for further improving the accuracy, robustness, and intelligence of fruit maturity identification.

Currently, research on tomato ripeness detection both domestically and internationally mainly focuses on color features, odor features, spectral features, and machine vision. For instance, Syahrir et al. [22] used RGB color images of tomatoes and transformed them into the color space format (L*a*b*). They then applied filtering and thresholding techniques to classify the tomato ripeness based on color. However, color features are susceptible to environmental influences, leading to limited accuracy in ripeness detection. Gómez et al. [23] utilized a specific electronic nose device (portable electronic nose, PEN 2) equipped with ten different metal oxide sensors. They combined principal component analysis (PCA) and linear discriminant analysis (LDA) to evaluate the volatile production changes related to tomato ripeness, aiming to differentiate different ripeness stages of tomatoes. However, the odor emitted by tomatoes at different ripeness stages can be affected by environmental and storage conditions, thereby influencing the electronic nose's detection results. Huang et al. [24] established a support vector machine discriminant analysis (SVMDA) model for each individual spatially resolved spectrum. They used spatially resolved spectral technology within the wavelength range of 550 to 1650 nm to assess six ripeness stages of tomatoes (i.e., green, breaker, turning, pink, light red, and red). Nevertheless, spectral feature recognition technology requires complex algorithms and models, and the instruments used are relatively expensive, making it difficult to implement on a large scale in practical production. On the other hand, Su et al. [25] conducted tomato ripeness classification using a lightweight YOLOv3 model and MobileNetV1 backbone network. The model can accurately recognize tomato ripeness even under the obstruction of leaves, and achieved a mAP (mean average precision) value of 97.5%. Li et al. [26] proposed a tomato ripeness recognition model called YOLOv5s, with an average precision of 97.42%. This improved model effectively addresses the problem of low recognition accuracy caused by obstructed small target tomatoes.

In summary, the deep learning method based on convolutional neural networks demonstrates clear advantages in fruit ripeness recognition. Applying a trained deep learning algorithm model to the mechanical harvesting equipment for cherry tomatoes enables high accuracy, as well as real-time and automated fruit recognition and ripeness detection, providing an intelligent solution for fruit picking processes [27]. This study focuses on the recognition and ripeness detection needs and challenges of cherry tomatoes in natural environments, and conducts a series of improvements on the YOLOv5n model. Our objective is to maintain model detection accuracy while minimizing the model's parameter quantity and computational resources to meet the requirements of the detection task. Through experiments and testing, the recognition effectiveness of cherry tomatoes in natural environments and the detection performance for ripe cherry tomatoes are evaluated, providing valuable references for the rational allocation of labor and precise target locking in the process of mechanized fruit harvesting.

## 2. Materials and Methods

### 2.1. Data Collection

Cherry tomato images were collected at the cherry tomato picking garden located at No. 308 Xingkang Road, Conghua District, Guangzhou City, Guangdong Province, China. The latitude and longitude of the location are 113.577071 and 23.609114, respectively. The images were captured on 7 March 2023, under natural daylight conditions, using a mobile phone camera at a shooting distance of approximately 10–60 cm. To mitigate the risk of overfitting the network model due to limited diversity in the training samples, the images were taken from left, right, and front angles. Two lighting conditions, representing different intensities of light, were considered. The images also encompassed various real-life growth postures of cherry tomatoes, including overlapping and adhering fruits. Figure 1

showcases the samples from the cherry tomato dataset captured in different scenarios. A total of 867 original cherry tomato images were collected and saved in .JPEG format with a resolution of 4000 × 3000 pixels. The acquired images underwent data cleaning and screening to remove low-quality pictures, such as those that were excessively blurry, severely overexposed, heavily occluded, or unrelated. This process resulted in a cherry fruit dataset consisting of 640 images.

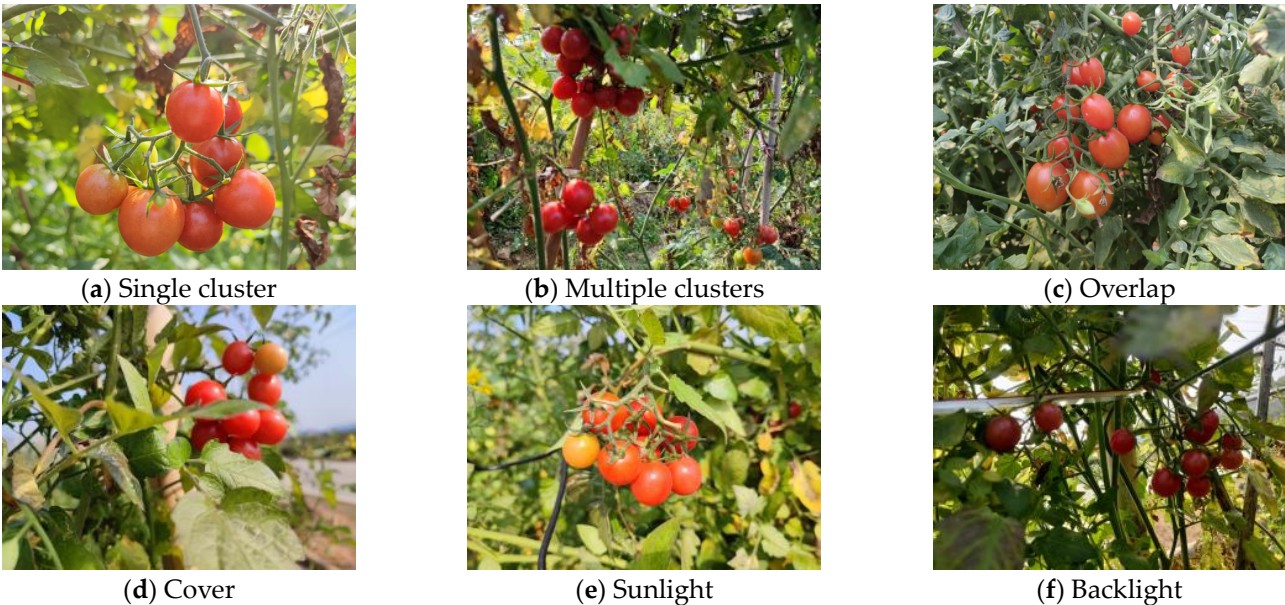

(**a**) Single cluster      (**b**) Multiple clusters      (**c**) Overlap

(**d**) Cover      (**e**) Sunlight      (**f**) Backlight

**Figure 1.** Cherry tomato samples from different scenarios.

*2.2. Data Processing*

In order to accurately identify the ripeness of cherry tomatoes in their natural environment, it is necessary to overcome the influence of external factors, such as obstructions from branches, leaves, and overlapping fruits. Cherry tomatoes ripen in different batches, and each batch may contain fruits at 1–3 different stages of ripeness, including mature, semi-mature, and immature fruits. Among these, the degree of coloring on the fruit's surface is the most crucial indicator of the cherry tomatoes' harvesting ripeness. Immature fruits have a color similar to that of branches and leaves, appearing greenish, while mature fruits display a bright red color. During the ripening process, the cherry tomatoes' fruit color changes gradually from greenish to bright red, with the degree of coloring becoming deeper, and the colored area expanding from small to large.

Based on China's national standard GH/T1193-2021 for tomatoes, cherry tomatoes are categorized into four periods: unripe, green ripe, color turning, and red ripe. In these periods, the pigmentation degree of immature and green ripe tomatoes is less than 40%, the pigmentation degree during the color turning period ranges from 40% to 70%, and the pigmentation degree of red ripe tomatoes ranges from 70% to 100%. In practical production, fruits are typically harvested during the color turning period. The use of algorithm recognition to obtain this information can help achieve accurate target positioning and reasonable allocation of labor, thus providing a reference basis for automatic fruit picking work.

In this study, cherry tomatoes with less than 40% peel color are referred to as unripe tomatoes, while those with more than 40% peel color are referred to as ripe tomatoes. Figure 2 provides examples of cherry tomatoes of different maturity. To reduce computational load without affecting the feature extraction of cherry tomatoes, the OpenCV "resize" function was employed to compress the image resolution to 1200 × 900 pixels. The LabelImg tool was used to manually annotate rectangular bounding boxes for cherry tomatoes in the dataset following the VOC annotation format. The annotation rules were as follows: For

cherry tomatoes that were completely exposed, a rectangular bounding box covering the entire fruit was annotated. For partially obscured or stuck-together cherry tomatoes, if the obscured portion was less than 90%, the exposed part was annotated with a rectangular bounding box. If a cherry tomato appeared partially outside the image boundary, or if more than 90% of the fruit was obscured, it was left unannotated. After completing the annotation process, we obtained a total of 640 images in XML format, containing the real ground truth annotations of the target objects. To make these annotations suitable for YOLO model training, we parsed the XML files in VOC format, extracted the target class and bounding box information, and converted this information into YOLO format in TXT files. Finally, we randomly split these TXT files according to a ratio of 7:1:2, resulting in 448 images for the training set, 64 images for the validation set, and 128 images for the test set.

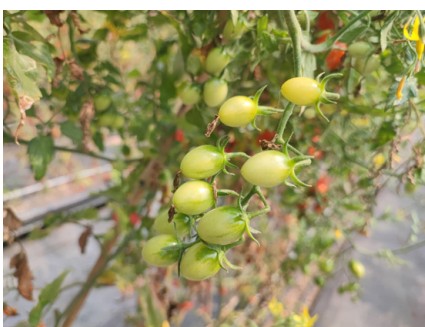 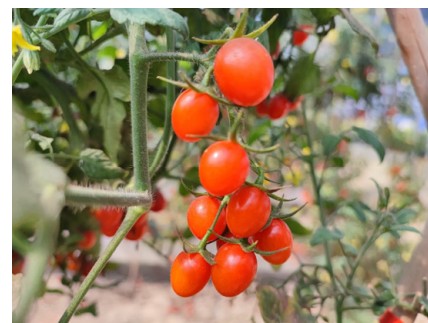

(**a**) Unripe cherry tomato          (**b**) Ripe cherry tomato

**Figure 2.** Examples of cherry tomatoes of different maturity.

## 3. Detection Method of Tomato Ripeness

### 3.1. Improved I-YOLOv5n Network Model Design

With the widespread application of computer vision and deep learning technology, target recognition algorithms based on deep convolutional neural networks (CNN) have become instrumental in the agricultural field. These algorithms can be categorized into two main types. The first type is a two-stage object detection algorithm that relies on region proposals, exemplified by the RCNN series [28], Fast R-CNN [29], and Faster R-CNN [30]. These algorithms generate candidate boxes for target regions, followed by the classification and positioning of these candidates, resulting in high detection accuracy. However, the multi-step process of these algorithms leads to relatively slower speeds. The second type is a one-stage object detection algorithm based on regression, which includes the YOLO series [31–36], SSD [37], and Efficient Det [38]. These algorithms directly output position and category information for objects within a single network. They demonstrate remarkable speed, enabling real-time object detection.

As a classic one-stage target detection algorithm, the YOLO series has gained widespread adoption due to its excellent detection performance. YOLOv5, developed and released by Ultralytics in June 2020, is faster and more accurate than its predecessors. The network architecture of the YOLOv5 model consists of input, backbone, neck, and detect modules. The input module receives three-channel RGB images with an input feature size of $640 \times 640 \times 3$. Various methods, like mosaic data augmentation, adaptive anchor frame calculation, and adaptive image scaling, are used for image preprocessing. The backbone module extracts features, such as edges, textures, and positions, from the input images, transforming them into multi-layer feature maps. The backbone network utilizes CSPDarknet53, which effectively improves feature extraction capability through residual structure and feature reuse mechanisms. Key structures within the backbone include the Conv module, C3 module, and SPPF module. The C3 module plays a crucial role in increasing network depth and receptive field, enhancing the model's feature extraction capabilities. The neck module connects the backbone and head modules, utilizing the feature pyramid

structure of FPN+PAN to fuse feature information from feature maps of different sizes. This improves the network's detection performance for targets of varying scales. The detect module predicts detection frames and categories, incorporating a boundary frame loss function and non-maximum suppression. The boundary frame loss utilizes the CIoU loss function [39], while non-maximum suppression filters out redundant detection frames of the same category, retaining high-confidence prediction frames.

The YOLO series models are characterized by complex network structures and numerous parameters, demanding significant GPU computing power for real-time detection. However, achieving real-time target recognition on embedded and mobile devices poses a notable challenge [40]. Therefore, further improvements are necessary to enable real-time target recognition on these devices. YOLOv5n is the smallest model in the YOLOv5 series, characterized by small model parameters and relatively low hardware requirements. It is particularly suitable for deployment on small embedded or mobile devices, although its accuracy is subpar. This article proposes an I-YOLOv5n network model based on YOLOv5n, aiming to enhance detection accuracy while preserving its smaller model size and faster detection speed. The improved structure is illustrated in Figure 3.

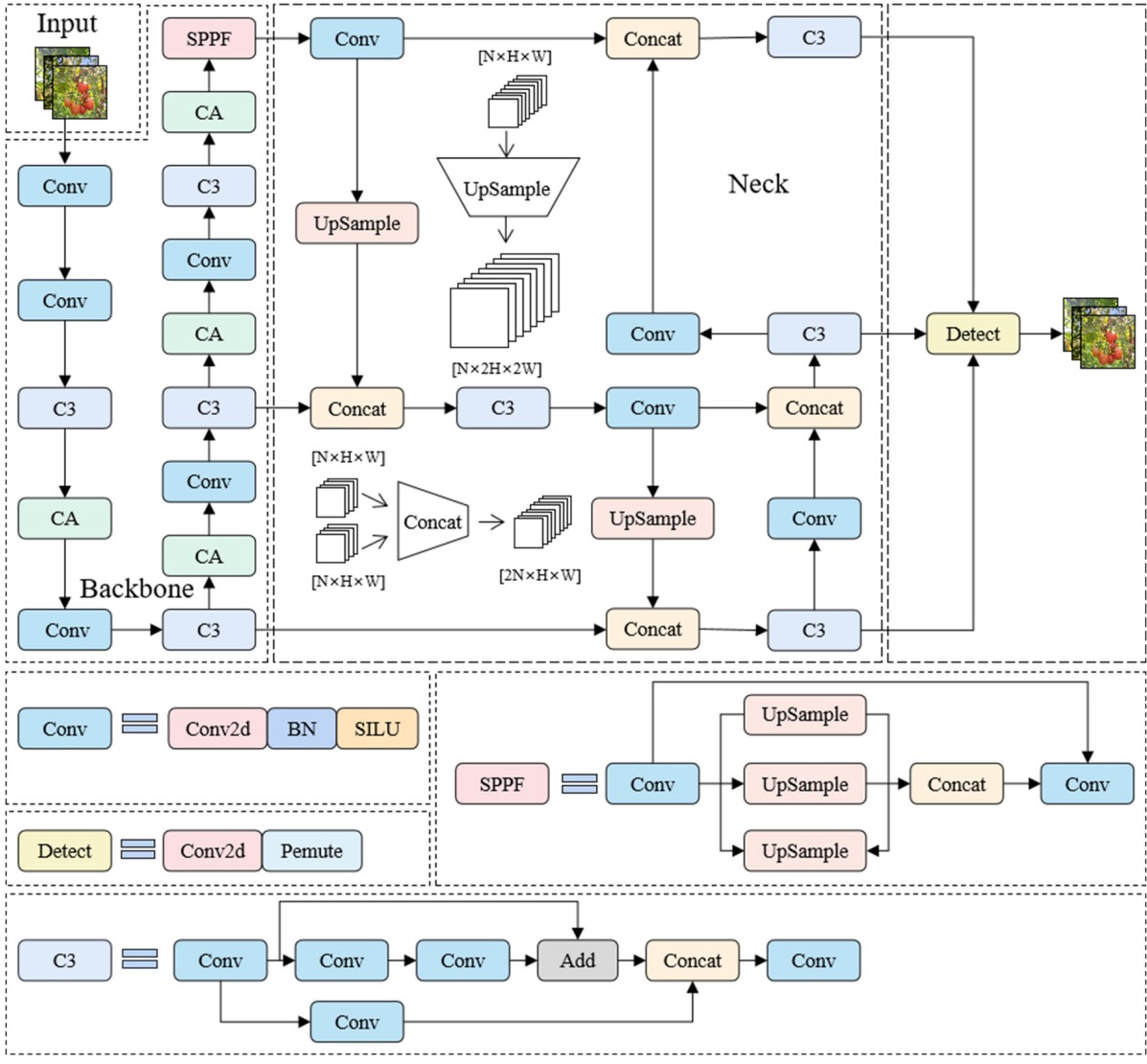

**Figure 3.** I-YOLOv5n network structure: Conv stands for the convolution operation, C3 is the feature extraction module, SPPF is the spatial pyramid pooling structure, UpSample is an upsampling operation, Concat is a feature fusion function, and CA represents the attention mechanism.

The I-YOLOv5n network model primarily focuses on three aspects of improvement: anchor frame clustering, attention mechanism, and loss function. Firstly, the K-means++ clustering algorithm [41] is employed to optimize the selection of anchor boxes. This approach ensures that the anchor boxes more accurately match the shape characteristics of cherry tomatoes, thereby enhancing the model's detection accuracy. Secondly, an attention module called CA [42] is added after the C3 module of the backbone network. This module expands the receptive field, filters out redundant and irrelevant feature channels, suppresses interference and noise in the model, and reduces the impact of complex backgrounds on target recognition. Finally, the traditional IOU loss function is replaced with the WIoU loss function [43]. Additionally, the outlier degree and dynamic nonmonotonic focusing mechanisms are introduced to address the boundary box regression balance issue between high-quality and low-quality data. These modifications collectively enhance the overall performance of the detector. The subsequent sections will delve into the underlying theories and improved methods associated with anchor frame clustering, attention mechanisms, and loss function in detail.

### 3.2. The k-Means++ Algorithm

Traditional target detection algorithms typically utilize a predefined set of anchor boxes to represent targets with varying scales and aspect ratios. These anchor boxes act as reference boxes during the training process, facilitating the generation of candidate boxes and the calculation of losses. However, the predetermined anchor box sizes are derived from clustering the Coco and VOC datasets. Given that the Coco dataset encompasses 80 different target types and the VOC dataset consists of 20 target types, each with varying sizes and categories, the anchor boxes obtained from clustering may not be entirely suitable for the current dataset. To address this issue, the YOLOv5 framework introduces the auto-learning bounding box anchors feature. This feature enables the model to autonomously learn and adjust anchor boxes that are better suited to the characteristics of the current dataset, employing the K-means clustering algorithm [44]. Throughout the training process, the model dynamically learns and updates the scale and aspect ratio of the anchor box, thereby enhancing the model's detection performance and generalization ability.

The main detection target of the object detection network in this study is the cherry tomato, which comes in various shapes and sizes. Figure 4 visualizes the anchor box size distribution and position distribution in the cherry tomato dataset used in this study. Figure 4a represents the distribution of the anchor box center points' position coordinates after normalizing the image resolution. The anchor box's position information is crucial for the model to locate the targets accurately. By providing accurate anchor box positions in the dataset, the model can learn the precise location of the targets in the images, leading to more accurate object detection and localization. From Figure 4a, it can be observed that the anchor box center points are uniformly distributed around the middle of the image, reflecting a relatively even spatial distribution of the targets in the dataset, without significant bias, and covering various locations effectively. Figure 4b shows the proportion of anchor box sizes relative to the image dimensions. The distribution of anchor box sizes reflects the range of target sizes in the images. A reasonable distribution of anchor box sizes ensures that the model can adapt to different-sized targets, achieving accurate detection and localization for various-sized objects. From Figure 4b, it is evident that the dataset contains targets of different sizes, with varying anchor box sizes and aspect ratios close to squares.

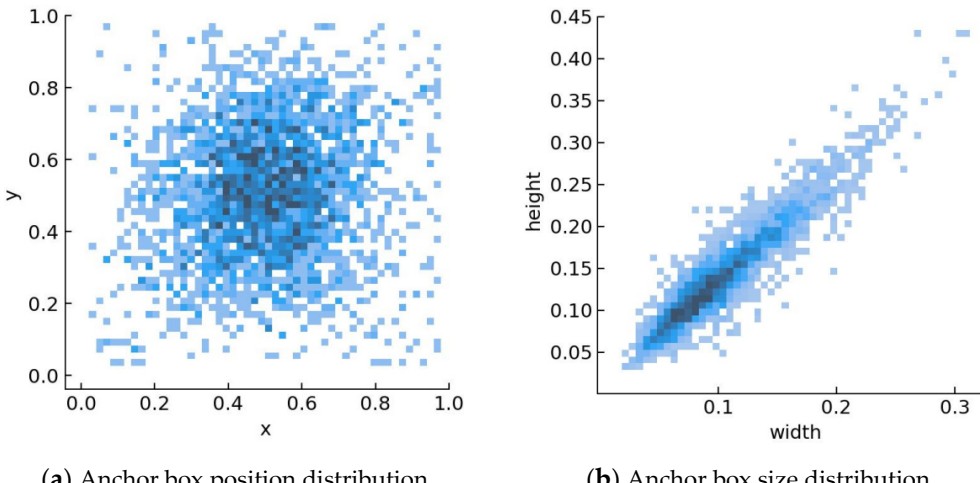

(**a**) Anchor box position distribution   (**b**) Anchor box size distribution

**Figure 4.** Distribution of anchor boxes in the datasets.

The auto-learning bounding box anchors feature in YOLOv5 utilizes the traditional K-means clustering algorithm to cluster anchor boxes in the dataset and obtain appropriate anchor boxes. However, a potential issue arises in the first step of the K-means clustering algorithm, which involves randomly selecting k cluster center points. This random selection can result in an uneven distribution of cluster centers, causing the algorithm to converge to local optima instead of the global minimum, and yielding incorrect clustering outcomes. On the other hand, the K-means++ clustering algorithm tends to select distant data points as the initial cluster center, enabling a better representation of the overall data structure. To overcome the drawbacks associated with initializing the cluster center, this article adopts an improved K-means++ algorithm for clustering the anchor boxes. The specific steps are as follows:

1.  Select a data point randomly from dataset $\chi$ as the initial clustering center, denoted as $C_1$.
2.  Compute the shortest distance $d(x)$ between every data point and the chosen cluster center $C_1$. Then, determine the next cluster center $C_2$ using the roulette algorithm, where its probability $p(x)$ is directly proportional to the square of the distance. The data point corresponding to the highest value of $p(x)$ will be selected as the subsequent cluster center $C_3$.

$$p(x) = \frac{D(x)^2}{\sum_{x \in \chi} D(x)^2} \tag{1}$$

3.  Repeating step 2, continue the process until k cluster centers have been selected. Assign the data points to their closest cluster center, and then update the cluster center's location based on the assigned data points.
4.  Continue to repeat steps 2 and 3 until either the cluster center remains unchanged or a predetermined number of iterations is reached.

After conducting a series of continuous iterative experiments, the nine anchor boxes ultimately selected for this study were (31,32), (43,46), (56,52), (54,65), (69,66), (78,81), (95,95), (117,115), (147,149).

### 3.3. Coordinate Attention Mechanism

The attention mechanism plays a crucial role in guiding the model to focus on pertinent "target" and "location" information. By incorporating attention mechanisms, the model gains the ability to dynamically learn the significance of each channel and enhance attention towards specific channels based on task requirements. This adaptive learning process filters out irrelevant information, thereby enhancing the model's representation ability and overall performance. To achieve more precise localization and identification of cherry tomatoes,

this study introduces the coordinate attention mechanism. This mechanism takes into account attention in both channel and spatial dimensions, enabling the model to allocate more attention to valuable channel information by learning adaptive channel weights. The specific principle behind this mechanism is illustrated in Figure 5.

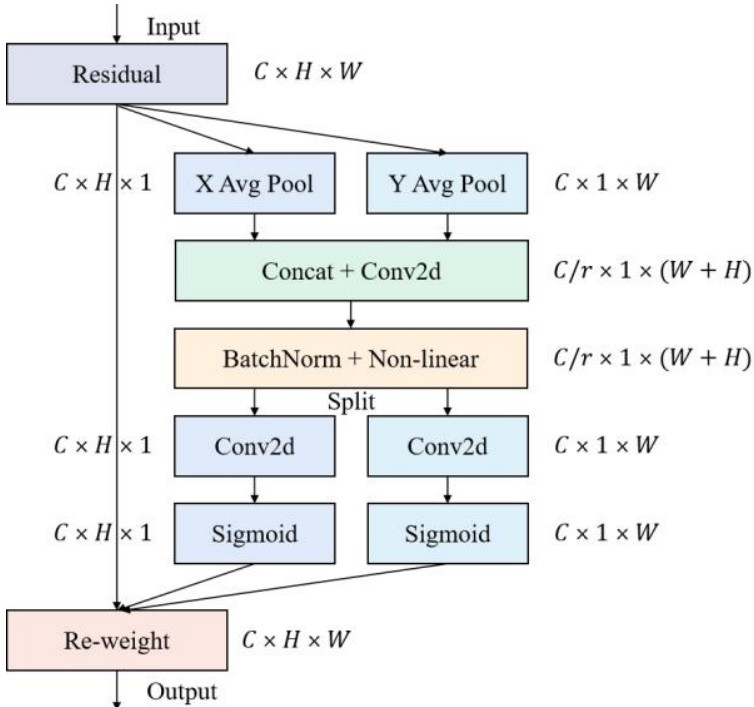

**Figure 5.** Coordinate attention mechanism. "X Avg Pool" and "Y Avg Pool", respectively, represent one-dimensional horizontal global pooling and one-dimensional vertical global pooling.

The coordinate attention mechanism encodes channel relationships and long-term dependencies by utilizing precise positional information. This mechanism is divided into two steps: coordinate information embedding and coordinate attention generation.

1. Coordinate Information Embedding

The two-dimensional global pooling method is commonly employed to globally encode spatial information through channel attention. However, the inclusion of position information within the channel attention mechanism in CA makes it challenging to preserve positional information. To mitigate the loss of positional information caused by two-dimensional global pooling, and to enable attention modules to capture precise positional information in remote spatial interactions, we propose decomposing the channel attention into two one-dimensional feature encoding processes. This approach effectively integrates spatial coordinates while addressing the issue of position information preservation.

For a given input X, each channel is encoded by applying pooling kernels of size (H, 1) or (1, W) along the horizontal and vertical coordinates, respectively. Consequently, we can express the output of the c-th channel with a height of h, as well as the output of the c-th channel with a width of w, as follows:

$$z_c^h(h) = \frac{1}{W} \sum_{0 \le i < W} x_c(h, i) \tag{2}$$

$$z_c^w(W) = \frac{1}{H} \sum_{0 \le j < H} x_c(j, w) \tag{3}$$

The aforementioned transformations serve to aggregate features along two spatial directions. Subsequently, these transformed feature maps are encoded into a pair of

attention maps that possess direction awareness and sensitivity to position. This enables the capturing of remote dependencies in one spatial direction while preserving precise position information in the other. By applying these attention maps in conjunction with the input feature maps, we can enhance the representation of objects of interest and achieve improved accuracy in locating and recognizing target regions.

2.    Coordinate Attention Generation

By incorporating coordinate information, we can effectively capture the global receptive field and encode precise position information. Connect Formulas (2) and (3), and send them to a shared $1 \times 1$ convolutional transformation $F_1$:

$$f = \delta\left(F_1\left(\left[z^h, z^w\right]\right)\right) \tag{4}$$

In the above equation, $\delta$ is the nonlinear activation function, $[\cdot, \cdot]$ is the concatenate operation along the spatial dimension, and f is the intermediate feature mapping encoding the spatial information in the horizontal and vertical directions.

Then, perform a split operation along the spatial dimension, using the other two $1 \times 1$ convolutional transformations $F_h$ and $F_w$, transform $f^h$ and $f^w$ into tensors with the same channel number as input X, and then use the sigmoid activation function to process them to obtain attention vectors $g^h$ and $g^w$.

$$g^h = \sigma\left(F_h\left(f^h\right)\right) \tag{5}$$

$$g^w = \sigma(F_w(f^w)) \tag{6}$$

Finally, extend $g^w$ and $g^h$, and the output formula of the coordinate attention Mechanism is

$$y_c(i,j) = x_c(i,j) \times g_c^h(i) \times g_c^w(j) \tag{7}$$

*3.4. Bounding Box Regression Loss with Dynamic Focusing Mechanism*

The loss function plays a crucial role in quantifying the discrepancy between the predicted information and the ground truth. A smaller value of the loss function indicates that the predicted information is closer to the actual information. In the case of YOLOV5, the loss function comprises three components: location loss, classification loss, and confidence loss. To acquire precise information about the location and maturity of cherry tomatoes in a natural environment, it is essential to optimize the loss function. This optimization aims to strike a balance in training errors related to prediction boxes, confidence scores, and classification. By fine-tuning the loss function, the model can achieve improved accuracy in detecting and classifying cherry tomatoes, thereby enhancing its overall performance in the natural environment.

This study introduces an enhanced approach to bounding box regression loss with dynamic focusing mechanism (WIoU) as the loss function. The WIoU method offers three versions: WIoU v1, which establishes an attention-based boundary box loss, and WIoU v2 and WIoU v3, which incorporate a focus mechanism through a gradient gain calculation method. In this paper, we adopt the WIoU v3 loss function, and compare it to the traditional intersection over union (IoU) loss function. WIoU v3 evaluates the anchor frame quality by leveraging the dynamic nonmonotonic focusing mechanism and outlier degree, addressing the challenge of achieving a balance in boundary box regression (BBR) between high-quality and low-quality data. This mechanism effectively reduces the influence of high-quality data competition and the adverse gradients generated by low-quality data. By dynamically allocating smaller gradient gains to low-quality anchor frames, the WIoU loss function enables BBR to prioritize processing an ordinary-quality anchor box, minimizing the impact of low-quality data on BBR. Consequently, the overall performance of the detector is significantly improved.

Figure 6 shows the relationship between anchor box and target box. From the figure, it can be concluded that the anchor box is $\vec{B} = [x\ y\ w\ h]$, and the target box is $\vec{B}_{gt} = \left[x_{gt}\ y_{gt}\ w_{gt}\ h_{gt}\right]$.

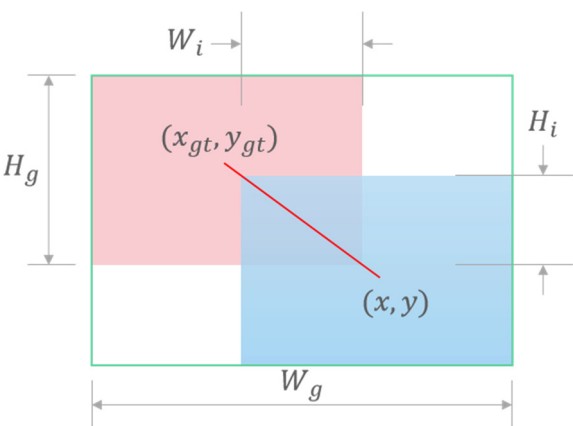

**Figure 6.** The relationship between anchor box and target box.

The IoU is used to measure the degree of overlap between the predicted bounding box and the actual bounding box, and the formula is

$$IoU = \frac{W_i H_i}{wh + w_{gt}h_{gt} - W_i H_i} \tag{8}$$

The IoU uses $L_{IoU}$ as the BBR loss and minimizes the gap between IoU by adjusting the position of the bounding box. The formula is

$$L_{IoU} = 1 - IoU \tag{9}$$

However, in cases where the predicted bounding box does not intersect with the actual bounding box, the IoU value becomes 0. This can lead to divergence during the training process, making it challenging to accurately assess the level of overlap between the predicted and actual bounding boxes. Currently, most loss functions assume that the training dataset consists of high-quality data, where the bounding box labels closely match the actual target locations. These loss functions primarily focus on improving the fitting capability of bounding box regression (BBR) losses. However, this assumption results in an overemphasis on training low-quality data during the training process. Geometric factors such as distance and aspect ratio amplify the impact of low-quality data, leading to unreasonable outcomes and a decline in the model's generalization performance. When the anchor box and target box have a strong overlap, the WIoU loss function reduces the penalty of geometric metrics. However, introducing multiple interventions during training can enhance the model's generalization ability. By doing so, the model becomes more robust and less sensitive to geometric variations, thereby improving its overall performance.

When the anchor box and target box overlap well, WIoU weakens the penalty of geometric metrics; based on this, we construct distance attention:

$$R_{WIoU} = \exp\left(\frac{(x - x_{gt})^2 + \left(y - y_{gt}\right)^2}{\left(W_g^2 + H_g^2\right)^*}\right) \tag{10}$$

In the formula, $R_{WIoU} \in [1,\ e)$, which will significantly amplify $L_{IoU}$ of the ordinary-quality anchor box.

The outlier degree of the anchor box is characterized by the ratio of $L_{IoU}$ to $\overline{L_{IoU}}$:

$$\beta = \frac{L_{IoU}^{*}}{\overline{L_{IoU}}} \in [0, +\infty] \tag{11}$$

As $\overline{L_{IoU}}$ is dynamic, the quality demarcation standard of anchor boxes is also dynamic. This enables WIoU v3 to dynamically allocate gradient gains, aligning with the current situation at every moment.

The quality of an anchor box is determined by its β value, where a smaller β indicates higher quality, while a larger β suggests lower quality. By assigning a small gradient gain to anchor boxes with smaller β values, the focus of boundary box regression can be directed towards ordinary-quality anchor boxes. Simultaneously, allocating a small gradient gain to anchor boxes with larger β values effectively prevents low-quality anchor boxes from generating larger harmful gradients. The formula for constructing the nonmonotonic focusing coefficient using β is as follows:

$$L_{WIoUv3} = rR_{WIoU}L_{IoU}, \ r = \frac{\beta}{\delta\alpha^{\beta-\delta}} \tag{12}$$

In the formula, $L_{IoU} \in [0, 1]$, which will significantly reduce $R_{WIoU}$ of the high-quality anchor box and its focus on the distance between central points when the anchor box coincides well with the target box.

By setting the initial value of $\overline{L_{IoU}} = 1$, we can prevent low-quality anchor boxes from lagging behind during the initial stages of training. This ensures that the anchor box obtained with $L_{IoU} = 1$ has the highest gradient gain. Additionally, to maintain this strategy during the early stages of training, we introduce a momentum factor m to delay the convergence of $\overline{L_{IoU}}$ towards $\overline{L_{IoU-resl}}$:

$$m = 1 - \sqrt[tn]{0.05} \tag{13}$$

This configuration facilitates the achievement of $\overline{L_{IoU}} \approx \overline{L_{IoU-resl}}$ after t-theory training. During the middle and later stages of training, WIoU v3 assigns small gradient gains to low-quality anchor boxes to minimize harmful gradients. Simultaneously, WIoU v3 prioritizes ordinary-quality anchor boxes to enhance the model's positioning performance.

### 4. Test Environment and Parameter Settings

*4.1. Test Platform*

In this study, the YOLOv5n network model was constructed and enhanced using PyCharm. The experiment and training were conducted on a Windows 10 operating system, utilizing a 12th Gen Intel (R) Core (TM) i5-12400F processor (Intel Corporation, Santa Clara, CA, USA) with a clock speed of 2.50 GHz, 16 GB of memory, and an NVIDIA GeForce RTX 3060 graphics card (NVIDIA Corporation, Santa Clara, CA, USA) with 8 GB of dedicated graphics memory. To accelerate operations, the CUDA 11.7 (NVIDIA, Santa Clara, CA, USA), cuDNN 8.5.0 (NVIDIA, Santa Clara, CA, USA), and Open CV 4.6.0 libraries (OpenCV, Menlo Park, CA, USA) were employed. The implementation of this study was carried out using the Python programming language (version 3.9.12) on the Keras deep learning framework.

*4.2. Network Training Parameter Settings*

The image input size is set to $640 \times 640 \times 3$. The training process consists of 1000 cycles with a batch size of 32 images. To accelerate the training of the neural network, the stochastic gradient descent (SGD) optimizer is utilized. The initial learning rate is set to 0.01, the momentum factor is 0.937, and the weight decay is 0.0005.

To enhance model performance and training efficiency, an early stopping strategy is employed. Real-time monitoring of the model's performance indicators takes place during

training, and early stopping is triggered when no improvement is observed in the last 100 training cycles. This strategy helps conserve computational resources.

Additionally, pretrained weight files and mosaic data augmentation functions are utilized to further improve the effectiveness of the model's training. These combined technologies and strategies contribute towards improved accuracy, generalization ability, and accelerated training.

### 4.3. Model Evaluation Indicators

When it comes to identifying the maturity of cherry tomatoes in a natural environment, it is crucial to take into account the accuracy and real-time performance of the detection network. To provide an objective measure of the model's detection effectiveness across different stages of cherry tomato maturity, this study employs several performance evaluation criteria, including precision (P), recall (R), average precision (AP), mean average precision (mAP), F1 score, and milliseconds (ms).

Precision refers to the proportion of correctly identified cherry tomatoes among the total number of cherry tomatoes predicted. Recall refers to the proportion of correctly identified cherry tomatoes among the actual number of cherry tomatoes. It is calculated using the following formula:

$$P = \frac{TP}{TP + FP} \tag{14}$$

$$R = \frac{TP}{TP + FN} \tag{15}$$

In the formula, TP is the number of correctly identified cherry tomatoes, FP is the number of misdetected cherry tomatoes, and FN is the number of undetected cherry tomatoes.

Average precision represents the area enclosed by a precision–recall (PR) curve, which is constructed using precision on the vertical axis and recall on the horizontal axis. It is calculated using the following formula:

$$AP = \int_0^1 P(R)\,dR \tag{16}$$

The average accuracy mAP is the average of all categories of APs used to evaluate the performance of the network model. The calculation formula is

$$mAP = \frac{1}{M} \sum_{k=1}^{M} AP(k) \tag{17}$$

In the formula, M is the total number of categories, and AP (k) is the k-th category AP value.

The F1 score is an evaluation metric commonly used to measure the accuracy of binary classification models. It can be regarded as a weighted average of the model's precision and recall, with a maximum value of 1 and a minimum value of 0. The F1 score is calculated using the following formula:

$$F1 = 2\frac{P \cdot R}{P + R} \tag{18}$$

The detection time refers to the average time taken by the target detection network to detect an image, typically measured in ms.

## 5. Experimental Results and Analysis
### 5.1. Improved I-YOLOv5 Object Detection Network

To validate the effectiveness of the improved network I-YOLOv5n based on YOLOv5n, we conducted a comparative analysis on the cherry tomato dataset before and after the enhancements. This analysis aimed to assess their performance differences in various

environments and for different maturity levels of cherry tomatoes. During the comparative analysis, we evaluated the accuracy, recall, average precision, and detection time of the cherry tomato detection. As illustrated in Table 1, I-YOLOv5n showcased remarkable advancements in accuracy and recall, surpassing YOLOv5n by 1.4 percentage points. Additionally, the average accuracy exhibited a boost of 0.3 percentage points, while the F1 score experienced a notable increase of 1.5 percentage points. It is important to note that despite the introduction of additional computations and parameters, I-YOLOv5n still outperformed YOLOv5n in terms of detection time. This indicates that the improved network structure of I-YOLOv5n enhances detection performance while maintaining a relatively fast detection speed.

**Table 1.** Comparison of test results of detection network before and after improvement.

| Network | P/% | R/% | mAP@0.5/% | F1/% | Test time/ms | Model Size (M) |
| --- | --- | --- | --- | --- | --- | --- |
| I-YOLOv5n | 95 | 92 | 95.2 | 93.5 | 5.3 | 4.4 |
| YOLOv5n | 93.6 | 90.6 | 94.9 | 92 | 6.5 | 3.9 |

Figure 7 showcases three selected samples, comparing the recognition results of the YOLOv5n network and the improved I-YOLOv5n network for different maturity levels of cherry tomatoes in diverse environments. These examples cover three lighting conditions: dim backlight, bright front light, and normal standard lighting. The chosen errors encompass four typical types: false detection, missed detection, duplicate recognition, and misclassification. In the figure, detection boxes labeled in red represent detected mature cherry tomatoes, detection boxes labeled in pink represent detected immature cherry tomatoes, and detection boxes labeled in yellow indicate manually annotated ground truths for the recognition results, highlighting the errors that occurred during algorithmic recognition.

The processing results of the original image and YOLOv5n are compared and analyzed as follows:

Sample 1: Recognition and detection of cherry tomatoes in a dim backlight environment. Two false detections were observed. The first false detection occurred when the algorithm incorrectly identified branches in the background as immature cherry tomatoes. The second error involved repeated identification and missed inspection of densely adhered cherry tomatoes.

Sample 2: Recognition and detection of cherry tomatoes in a bright smooth environment. A single false detection was observed, where the algorithm mistakenly identified the leaves in the background as immature cherry tomatoes.

Sample 3: Recognition and detection of cherry tomatoes under normal standard lighting conditions. Two false detections were identified. The first error involved a category recognition mistake, where immature cherry tomatoes were incorrectly identified as mature cherry tomatoes. The second error occurred when the algorithm identified branches in the background as mature cherry tomatoes.

Through the analysis of these errors, it can be observed that in YOLOv5n, the majority of errors were caused by occlusion from branches and environmental interference. To address these issues, we made improvements to the original YOLOv5n model. Firstly, we introduced CA (context aggregation) to enhance the model's perception and understanding of key information by removing background interference. This mechanism helps the model accurately locate and identify the target regions. Secondly, we employed the K-means++ algorithm to re-cluster the three sets of anchor boxes provided by YOLOv5n. By generating anchor boxes that better match the target sizes and aspect ratios based on the real distribution of objects in the dataset, the model can more comprehensively search for target objects, reducing instances of missed detection. Additionally, we replaced the traditional loss function with the WIoU (weighted intersection over union) loss function to address the imbalance and weighting issues in the dataset. This allows for more effective training and optimization of the model, improving its performance and generalization

ability. These improvements enable I-YOLOv5n to handle occlusion and environmental interference more effectively, enhancing its detection capability for cherry tomatoes of different scales and maturity levels.

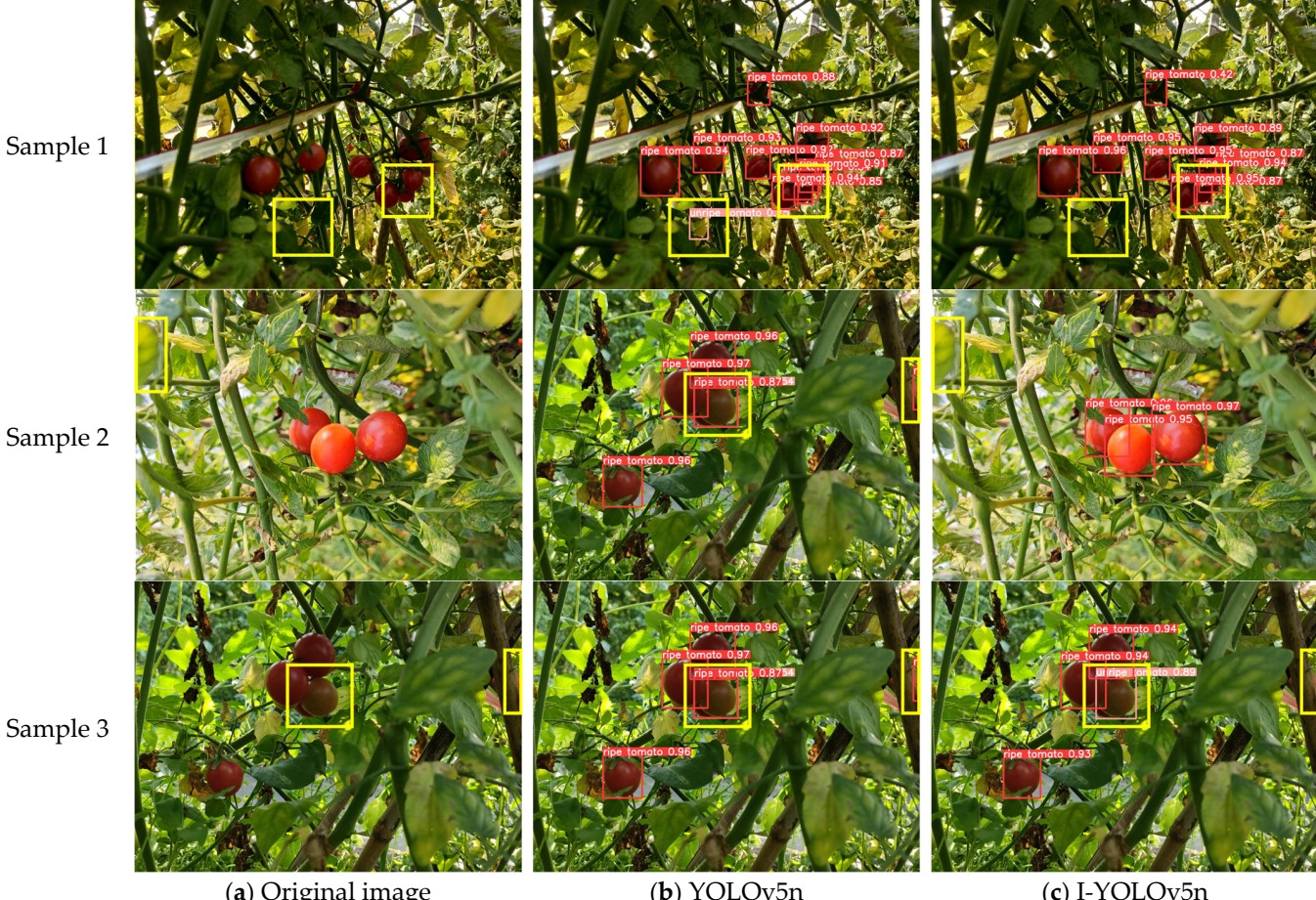

**(a)** Original image        **(b)** YOLOv5n        **(c)** I-YOLOv5n

**Figure 7.** Comparison of detection network recognition results before and after improvement.

By comparing the processing results of the original image, YOLOv5n, and the improved I-YOLOv5n, it becomes evident that the improved I-YOLOv5n outperforms in terms of reducing both false positives and false negatives. It demonstrates the ability to accurately identify target areas, thereby enhancing the accuracy and recall of target detection and improving the overall performance of the model. This indicates that the improved I-YOLOv5n exhibits higher accuracy and robustness in object detection tasks, positioning it as a superior model.

*5.2. Ablation Experiments*

In the ablation experiment, we employed the YOOv5n model as the baseline model and introduced the K-means++ algorithm, CA attention module, and WIoU loss function as enhancements. Comparing these improvements with the original model, the three proposed enhancements had positive impacts on various aspects of the model. The experimental analysis of these improvement measures is presented in Table 2.

**Table 2.** Comparison of ablation experiment performance.

| Network | P/% | R/% | mAP@0.5 /% | F1/% | Model Size (M) |
|---|---|---|---|---|---|
| YOLOv5n | 93.6 | 90.6 | 94.9 | 92 | 3.9 |
| + K-means++ | 96.1 | 88.7 | 95.3 | 92.3 | 3.9 |
| + CA | 95.2 | 90.1 | 95 | 92.6 | 4.4 |
| + WIoU | 97 | 88 | 94.7 | 92.3 | 3.9 |
| + K-means++ + CA | 94.2 | 90 | 94.3 | 92.1 | 4.4 |
| + CA+ WIoU | 92.7 | 91.3 | 94.6 | 92 | 4.4 |
| + K-means++ + CA+ WIoU | 95 | 92 | 95.2 | 93.5 | 4.4 |

Firstly, we introduced an enhanced K-means++ algorithm to improve the alignment between anchor boxes and target objects. This enhancement resulted in a significant increase in model accuracy and F1 score by 2.5 percentage points and 0.4 percentage points, respectively. In Figure 8, we provide a comparison between the YOLOv5n model before and after applying the K-means++ algorithm improvement. We have highlighted the areas where the anchor box exhibits notable changes. It is evident that the re-clustering of anchor boxes better conforms to the true distribution of cherry tomatoes. This enables the model to capture more accurate size, position, and shape information of the cherry tomatoes, leading to improved correspondence with the target object. As a result, the detection accuracy and positioning accuracy of the target are enhanced.

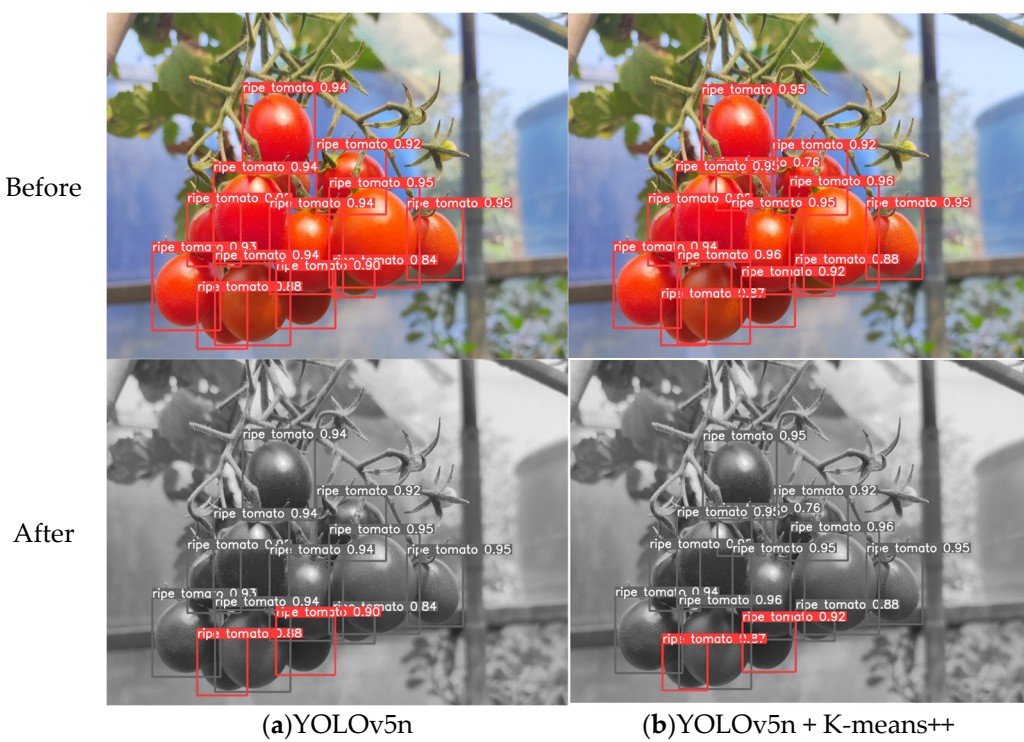

(**a**)YOLOv5n     (**b**)YOLOv5n + K-means++

**Figure 8.** Comparison before and after improving K-means++algorithm.

Secondly, the incorporation of the CA attention module resulted in a 1.6 percentage point improvement in model accuracy. The CA module dynamically learns the significance of each channel and enhances attention towards specific channels. In Figure 9, we present a comparison between the YOLOv5n model before and after the integration of the CA attention module. It is evident that the introduction of the CA attention module has led to an average increase of 0.03 points in confidence for recognizing unobstructed cherry tomatoes. Additionally, the module enables accurate recognition of semi-occluded cherry tomatoes that were previously unrecognizable. By focusing on essential feature channels,

the model can effectively filter out irrelevant information and better extract cherry tomato features, thereby enhancing its ability to recognize and locate cherry tomatoes.

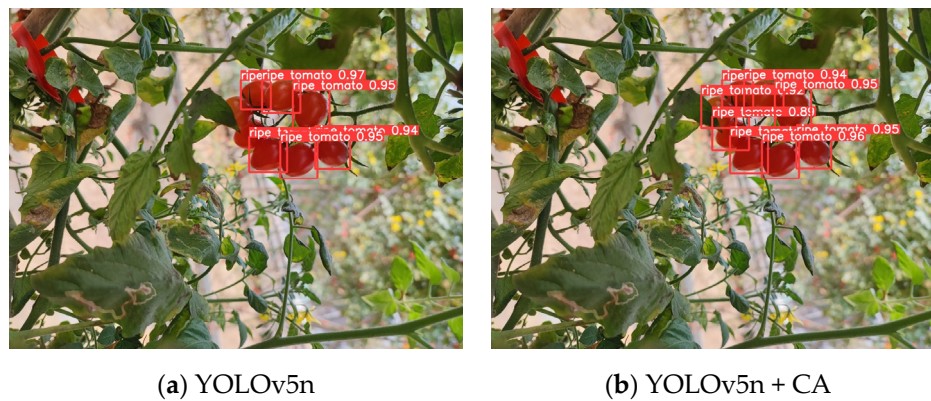

(**a**) YOLOv5n          (**b**) YOLOv5n + CA

**Figure 9.** Comparison before and after improving CA mechanism.

Finally, the integration of the WIoU loss function resulted in a significant improvement in model accuracy by 3.4 percentage points. By adjusting the gradient gain allocation strategy, the WIoU loss function directs the model's attention towards ordinary-quality training samples while reducing the harmful gradients generated by low-quality data. This enables the model to learn the characteristics of the target object more effectively and consistently. Evaluating the effectiveness of model improvements involves analyzing the loss curves during the training process. In Figure 10, we compare the loss curves before and after modifying the loss function of the YOLOv5n model. It is evident that the introduction of the WIoU loss function accelerates the fitting process, indicating that the model can better capture the characteristics of the target object, thereby enhancing overall accuracy.

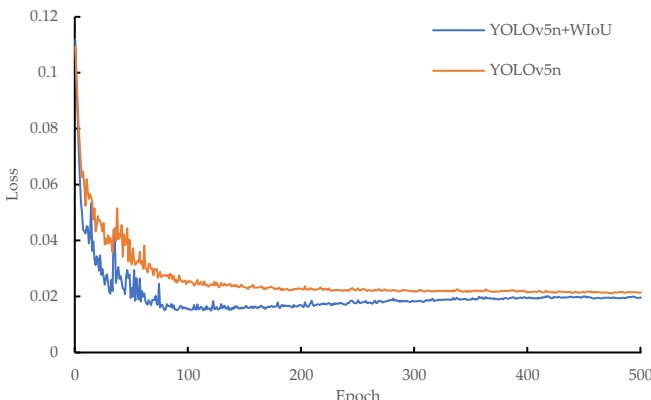

**Figure 10.** The loss curve of the training process before and after modifying the loss function.

### 5.3. Comparison of Different Attention Mechanisms

Attention mechanism plays an important role in computer vision tasks, as it can help models focus on important information in images and improve model performance. This article compares five common attention mechanisms: CA, the convolutional block attention module (CBAM) [45], efficient channel attention (ECA) [46], squeeze and excitation (SE) networks [47], and the similarity-based attention module (Sim AM) [48]. These attention mechanisms are evaluated experimentally using the YOLOv5n model. The results are shown in Table 3. Among them, CA captures the dependency relationship between channels by modeling the correlation between channels in the feature map. CBAM is an attention mechanism that combines spatial and channel attention. ECA introduces an efficient channel attention module for extracting correlation information between channels. SE adaptively adjusts the importance of channels by introducing compression and excitation

operations. The Sim AM determines the attention weight of each position by calculating the similarity between different positions in the feature map.

**Table 3.** Comparison of the attention mechanism of the model.

| Attention Mechanism | P/% | R/% | mAP@0.5/% | Layers | Parameters | Model Size (MB) |
|---|---|---|---|---|---|---|
| CA | 95.2 | 90.1 | 95 | 253 | 2,025,391 | 4.4 |
| CBAM | 94.3 | 90.9 | 94.5 | 257 | 1,936,343 | 4.2 |
| ECA | 93.5 | 90.5 | 94.7 | 229 | 1,761,883 | 3.9 |
| SE | 94.6 | 88.7 | 94.6 | 237 | 1,935,951 | 4.2 |
| Sim AM | 94.5 | 91.6 | 95.2 | 221 | 1,761,871 | 3.9 |

After conducting detailed data analysis and comparison, the model incorporating the CA module demonstrated outstanding performance in terms of accuracy, achieving an impressive 95.2%. Compared to the CBAM, ECA, SE, and Sim AM attention modules, the CA module exhibited significant advantages, with accuracy improvements of 0.9, 1.7, 0.6, and 0.7 percentage points, respectively. Moreover, the CA module also exhibited strong performance in terms of mAP indicators, surpassing CBAM, ECA, and SE by 0.5, 0.3, and 0.4 percentage points, respectively, although its performance was slightly lower than that of Sim AM. Furthermore, despite the slightly larger weight file size of the CA module compared to the other four modules, there was no significant increase in parameters or computational complexity. This implies that the CA module can be effectively utilized to enhance the accuracy of cherry tomato detection without compromising overall performance.

In summary, based on the comprehensive analysis and practical considerations, the CA module proves to be the most suitable choice for the cherry tomato detection task, delivering remarkable accuracy improvements without introducing excessive complexity.

*5.4. Comparison with Other Deep Learning Models*

Based on the analysis of existing target detection algorithms, this research places particular emphasis on real-time cherry tomato detection. Consequently, YOLO series models have been selected due to their speed and suitability for real-time applications. This article conducts a comprehensive comparison between the proposed I-YOLOv5n model and other popular models such as YOLOv5n, YOLOv5s, YOLOv7, YOLOv8n, and YOLOv8s. The evaluation metrics include precision, recall, F1 score, mAP, test time, and weight file size using the optimal weight file. The results of these evaluations are presented in Table 4.

**Table 4.** Comparison of detection performance of different detection networks.

| Model | P/% | R/% | mAP@0.5/% | F1/% | Test Time/ms | Model Size (MB) |
|---|---|---|---|---|---|---|
| I-YOLOv5n | 95.0 | 92.0 | 95.2 | 93.5 | 5.3 | 4.4 |
| YOLOv5n | 93.6 | 90.6 | 94.9 | 92.0 | 6.5 | 3.9 |
| YOLOv5s | 94.6 | 89.5 | 94.7 | 91.9 | 7.1 | 14.4 |
| YOLOv7 | 94.3 | 88.9 | 94.0 | 91.8 | 27.8 | 74.8 |
| YOLOv8n | 93.8 | 88.7 | 94.5 | 91.2 | 6.4 | 6.2 |
| YOLOv8s | 94.1 | 90.4 | 95.1 | 92.0 | 8.1 | 22.5 |

Considering that the model will be used for embedded and mobile devices in the development of an unmanned cherry tomato mechanical picking system, the model size and test time of the model become one of the main considerations. After evaluating multiple models, we found that YOLOv5n has the smallest model size, only 3.9 MB, and a test time of 6.5 ms. In contrast, YOLOv7 has the largest model size, 19 times that of YOLOv5n, reaching 74.8 MB, and it also has the longest test time among the five models, which is 27.8 ms. YOLOv5s and YOLOv8s have relatively larger model sizes, 14.4 MB and

22.5 MB, respectively, with test times of 7.1 ms and 8.1 ms, slightly slower than YOLOv5n. As for YOLOv8n, its model size is 6.2 MB, its test time is 6.4 ms, which is almost the same as YOLOv5n, but its recall rate has dropped by 1.9 percentage points. Taking all these factors into consideration, we chose YOLOv5n as the baseline model because of its superior performance in model size and test time. Its lower computational requirements make it very suitable for real-time applications and resource-constrained scenarios on embedded and mobile devices. The improved I-YOLOv5n has a model size of 4.4 MB, only 0.5 MB larger than YOLOv5n, and the computational load remains within an acceptable range. The test time of this model significantly improved to 5.3 ms, surpassing YOLOv5n, YOLOv5s, YOLOv7, YOLOv8n, and YOLOv8s by 1.22, 1.34, 5.25, 1.2, and 1.53 times, respectively. The model's precision increased to 95%, exceeding YOLOv5n, YOLOv5s, YOLOv7, YOLOv8n, and YOLOv8s by 1.4, 0.4, 0.7, 1.2, and 0.9 percentage points, respectively. The recall also improved to 92%, which is 1.4, 2.5, 3.1, 3.3, and 1.6 percentage points higher than YOLOv5n, YOLOv5s, YOLOv7, YOLOv8n, and YOLOv8s, respectively.

To sum up, the improved I-YOLOv5n model achieves a remarkable test time of 5.3 ms while maintaining a 95.2% mAP, with a model size of only 4.4 MB. It offers a very fast computation speed and provides high accuracy in object recognition. Therefore, this model is highly suitable for embedded and mobile devices in the development of an unmanned cherry tomato picking system.

Figure 11 compares the confusion matrices of different detection networks. Each confusion matrix is presented in a tabular form, and divides the predicted results into four different categories: true positive (TP), true negative (TN), false positive (FP), and false negative (FN). The horizontal axis represents the true labels (actual categories), while the vertical axis represents the predicted results (predicted categories). The color intensity of the matrix entries indicates the quantity or proportion of the corresponding entries. Generally, darker colors represent a higher quantity or proportion, while lighter colors represent a lower quantity or proportion. This visualization scheme helps us observe the differences between different categories and the prediction trends of the model.

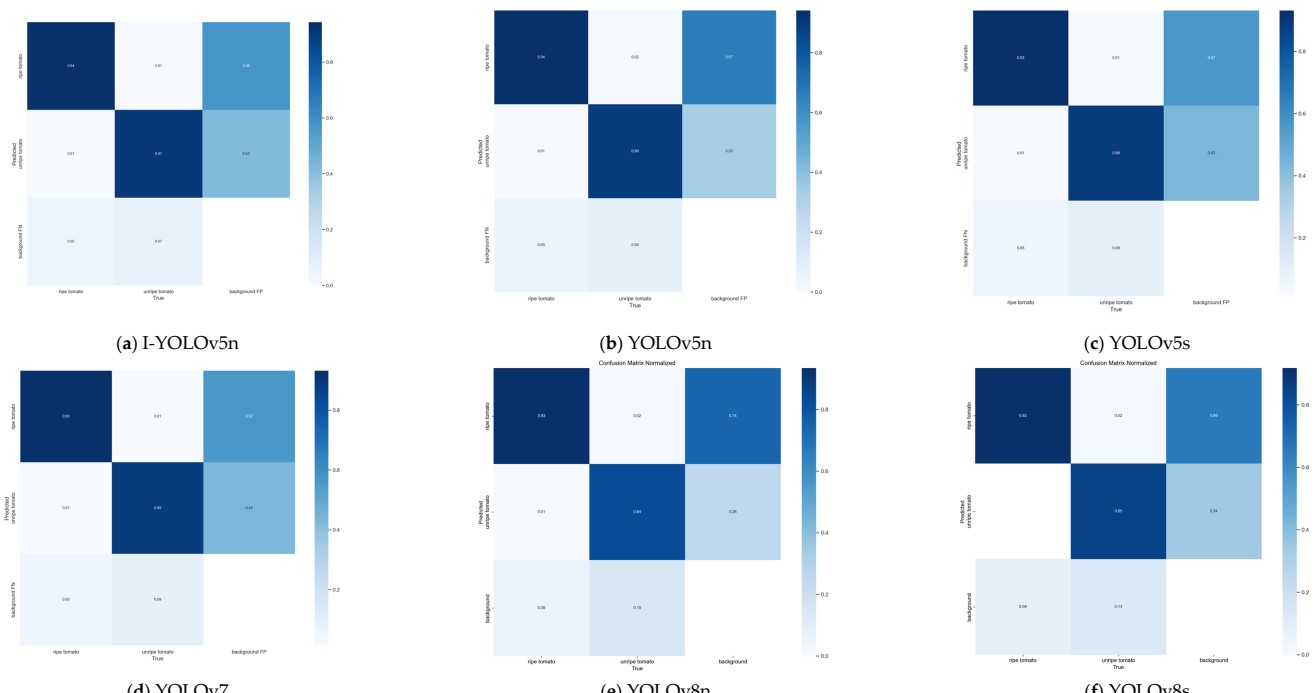

**Figure 11.** Comparison of confusion matrices of different detection networks.

Based on the analysis of the confusion matrices, it can be observed that there are relatively few false detections between the two maturity categories of cherry tomatoes, with the false detection rate concentrated between 0 and 0.02. This can be attributed to the distinct color differences between mature and immature cherry tomatoes, enabling the model to accurately distinguish between them. However, due to the similarity in color between immature cherry tomatoes and the branches and leaves in the background, some immature tomatoes were mistakenly classified as background, resulting in a false detection rate of 0.07–0.15. Among the tested models, the I-YOLOv5n model in Figure 11a has the lowest error detection rate, and the TPs of mature cherry tomatoes and immature cherry tomatoes reached 0.94 and 0.91, respectively, which are the highest values in the comparison model.

In Figure 12, we visually analyze and compare the recognition results of different detection networks. In the figure, detection boxes labeled in red represent detected mature cherry tomatoes, detection boxes labeled in pink represent detected immature cherry tomatoes, and detection boxes labeled in yellow indicate manually annotated ground truths for the recognition results, highlighting the errors that occurred during algorithmic recognition. In Figure 12a, YOLOv5n, YOLOv5s, YOLOv7, and YOLOv8n mistakenly identified the half leaf on the left side of the image as an immature cherry tomato. In Figure 12b, YOLOv5n and YOLOv5s missed one immature cherry tomato, while YOLOv8n and YOLOv8s missed two immature cherry tomatoes. In Figure 12c, YOLOv5s mistakenly detected two cherry tomatoes, while YOLOv5n, YOLOv7, YOLOv8s, and YOLOv8n also made incorrect detections. However, the I-YOLOv5n model accurately identified both the false positives and missed detections mentioned above.

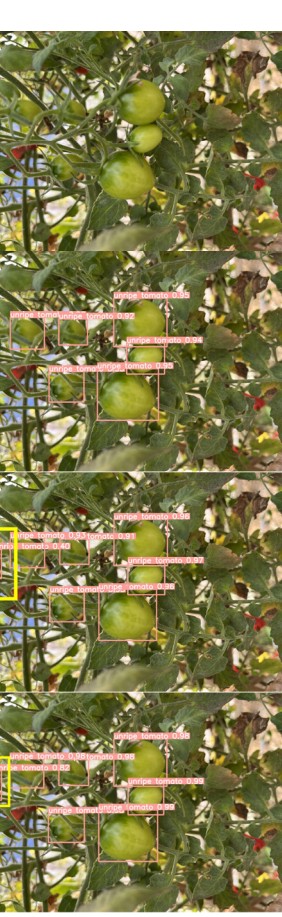 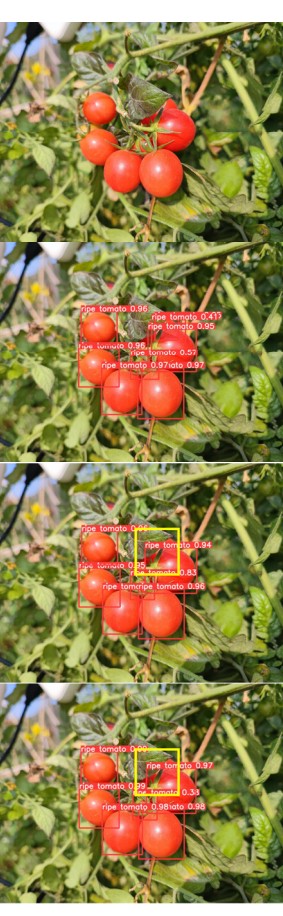 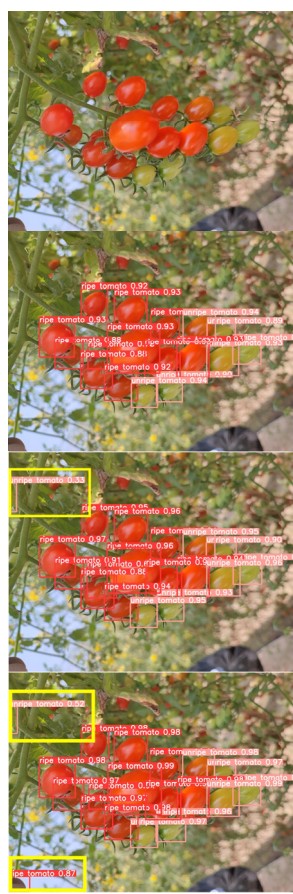

**Figure 12.** *Cont.*

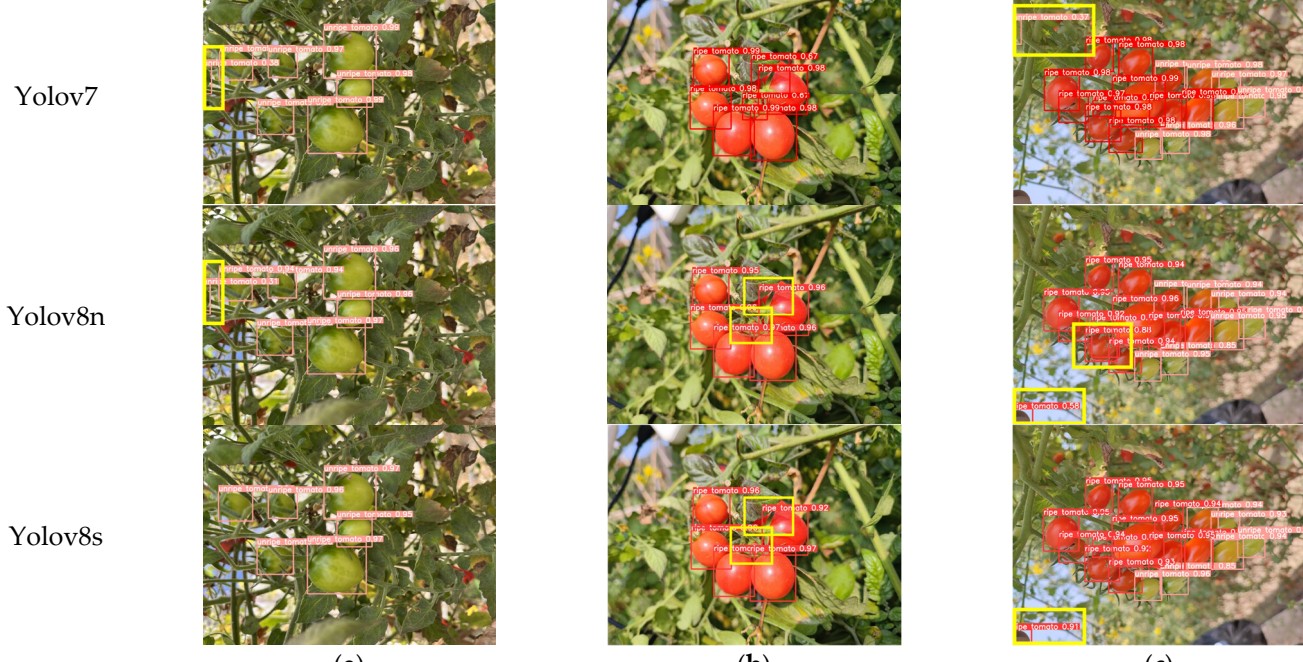

Yolov7

Yolov8n

Yolov8s

(**a**)      (**b**)      (**c**)

**Figure 12.** Comparison of recognition results of different detection networks. In (**a**–**c**), we can observe images of immature cherry tomatoes, mature cherry tomatoes, and mixed cherry tomatoes with varying degrees of maturity, respectively. The first row displays the original images, while the second to seventh rows depict the results obtained using the I-YOLOv5n, YOLOv5n, YOLOv5s, YOLOv7, YOLOv8n, and YOLOv8s models, respectively.

Through comprehensive comparison, it is evident that the I-YOLOv5n model demonstrates accurate identification of immature and mature cherry tomatoes under various conditions, including different quantities, sparsity, branches and leaves, and maturity levels. Compared to the other five object detection networks, the I-YOLOv5n model exhibits higher recognition accuracy with fewer false positives and missed detections. This demonstrates the strong robustness of the algorithm proposed in this study, making it adaptable to various scenarios in different natural environments.

## 6. Conclusions

This research presents an improved YOLOv5n model, named I-YOLOv5n, for cherry tomato maturity recognition and target location detection. The model incorporates the CA module after the C3 module of the backbone to reduce the interference of complex backgrounds on target recognition. It utilizes the WIoU loss function, along with the outlier degree and dynamic nonmonotonic focusing mechanism, to address the challenge of balancing boundary box regression between high-quality and low-quality data, thus enhancing the overall performance of the detector. The model occupies only 4.4 MB of memory, making it suitable for real-time applications and accurate target positioning and maturity grading in the mechanical automatic picking of cherry tomatoes. This research provides valuable insights for the field of agricultural automation, and holds promising application prospects.

The present research aimed to develop a specialized dataset tailored specifically for cherry tomato images captured in natural environments. This dataset was used to assess the performance of the I-YOLOv5n model. The experimental results demonstrate that the model achieves an average accuracy of 95%, a recall rate of 92%, a mAP of 95.2%, and a detection speed of 5.3 ms, showcasing superior recognition accuracy and speed performance.

In future work, we will further improve the dataset for cherry tomatoes and build a more refined classification maturity detection method to meet the needs of mechanized harvesting of cherry tomatoes in more complex environments. We will also strive to enhance the robustness of the model, combining robotic arms and depth cameras, to apply the model to more application fields.

**Author Contributions:** Conceptualization, C.W. (Congyue Wang) and Y.L. (Yubin Lan); Formal analysis, L.W., J.W. and J.L.; Funding acquisition, Y.L. (Yubin Lan); Investigation, C.W. (Congyue Wang) and C.W. (Chaofeng Wang); Methodology, C.W. (Congyue Wang), C.W. (Chaofeng Wang) and Y.L. (Yubin Lan); Software, J.L.; Validation, C.W. (Congyue Wang) and C.W. (Chaofeng Wang); Visualization, L.W., J.W. and J.L.; Writing—original draft, C.W. (Congyue Wang) and C.W. (Chaofeng Wang); Writing—review and editing, C.W. (Congyue Wang) and Y.L. (Yuanhong Li). All authors have read and agreed to the published version of the manuscript.

**Funding:** The study was funded by the Laboratory of Lingnan Modern Agriculture Project (NT2021009), the leading talents program of Guangdong Province (2016LJ06G689), the 111 Project (D18019), Guangdong Basic and Applied Basic Research Foundation (2021A1515110554), the Key-Area Research and Development Program of Guangdong Province (2019B020214003), the China Postdoctoral Science Foundation (2022M721201), and the open competition program of the top ten critical priorities of Agricultural Science and Technology Innovation for the 14th Five-Year Plan of Guangdong Province (2022SDZG03).

**Data Availability Statement:** The data presented in this study are available on request from the corresponding author.

**Acknowledgments:** We would also like to thank all reviewers for their valuable comments.

**Conflicts of Interest:** The authors declare no conflict of interest.

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
