# Peer review of "A Lightweight Cherry Tomato Maturity Real-Time Detection Algorithm Based on Improved YOLOV5n"

_agronomy, doi:10.3390/agronomy13082106_

Round 1

Reviewer 1 Report

In this paper, the Authors present A Lightweight Cherry Tomato Maturity Real Time Detection Algorithm Based on Improved YOLOV5n.

After a detailed review, it's a very goog quality paper with exclellent experimental results and analysis, but I also consider it needs to improve some aspects. For a more detailed explanation, please, see my comments below.

1. The authors describe the motivation and the problem well. But, during the description of the scientific background, the literature review is just a pile of information, lacking of analysis and induction.

2. The authors are invited to check the title of figure 1, 7 and 9. They must be in the same page. 

3. The authors should chnage the tiltle number (5.1,5.2, etc.) after Experimental results and Analysis section

4. We suggest adding the link for giving an opportunity to the scientific community to access the data easily

5. In addition, you'd better give some more details about data augmentation such as the total number of images after applied this process.

Regards

Reviewer 2 Report

This paper describes a YOLOv5-based approach for assessing cherry tomato maturity. The paper is well written. However, I have the following considerations.

1.       The Introduction section needs to include some relevant recent works in tomato detection. Please improve this section by adding the following relevant citations:

a.       https://doi.org/10.1016/j.compag.2023.107757

b.       https://doi.org/10.1016/j.compag.2022.106780

c.       https://doi.org/10.1007/s11042-021-10933-w

2.       YOLOv5 is not compatible with the VOC format. Since the authors provided an extremely detailed procedure in section 2.2, they can add a note stating they converted data from the VOC format returned by LabelImg into a proper format.

3.       Other works in the literature already use one or more of the modifications proposed by the authors. Hence, the overall innovation in the model must be stressed.

4.       The importance of Figure 4a should be further stressed.

5.       From the results reported in Table 4, it seems that the proposed architecture has only marginal improvements with respect to the original one. Furthermore, the model size is comparable, and I honestly don’t think that a difference of 10-15 MBs for a modern device is relevant, even if the device is extremely constrained. Hence, I think the authors should focus their discussion on the test speed improvements and evaluate whether similar improvements can be applied to the newer version of YOLO (YOLOv8n).

For this, I suggest a major revision. I also suggest checking the narrative for typos and readability. 

Minor typos and readability issues have been found throughout the paper. Please double-check and proofread it.

Round 2

Reviewer 2 Report

The authors addressed only a part of the highlighted issues. Still, due to the quality of the work, the editor can consider the paper for publication.